# Investigation of Underwater Shoulder Muscle Activity during Manikin-Carrying in Young Elite Lifesaving Athletes

**DOI:** 10.3390/s22062143

**Published:** 2022-03-10

**Authors:** Daniel Hon-Ting Tse, Wan-Yu Kwok, Billy Chun-Lung So

**Affiliations:** Gait and Motion Analysis Laboratory, Department of Rehabilitation Sciences, The Hong Kong Polytechnic University, Hong Kong, China; httse@polyu.edu.hk (D.H.-T.T.); wan-yu-aryu.kwok@connect.polyu.hk (W.-Y.K.)

**Keywords:** anthropometry, shoulder, surface electromyography, waterproof, sport lifesaving, muscle activation

## Abstract

Manikin carrying is a lifesaving sports technique, in which athletes stroke with one arm and carry a manikin of 60 kg with the other arm as they swim. Stabilizing the manikin exerts great demand on the shoulder muscles of the carrying arm; thus, this study aimed to investigate the muscle activation of the carrying shoulder and the possible factors associated with it. This was a cross-sectional study, in which 20 young elite lifesaving athletes were recruited from the Hong Kong Lifesaving Society. The muscle activity of the posterior deltoid (PD), teres major (TM), and middle trapezius (MT) were recorded with wireless surface electromyography (sEMG) during the performance of 25-m manikin carrying in a swimming pool. The 25-m manikin-carrying was divided into and analyzed in 3 phases: initial, middle, and end phase. The initial phase was defined as the period from the athlete’s first swimming stroke to the end of the third stroke; the middle phase was defined as the period between the initial and the end phase; and the end phase was defined as the period from the last third stroke to the last stroke at the 25-m finishing line. The first web space and grip strength were measured. The speed and number of inhalations were calculated. PD showed muscle activity of 55.73% of maximal voluntary isometric contraction (MVIC) in the initial phase and 40.21% MVIC in middle phase. TM showed a muscle activity of 65.26% MVIC in the initial phase and 64.35% MVIC in the middle phase. MT showed 84.54% MVIC in the initial phase and 68.54% MVIC in the middle phase. Young elite athletes showed significant use of PD, TM, and MT during manikin-carrying. The muscle activity levels correlated with the first web space, grip strength, speed, and number of inhalations of the athletes.

## 1. Introduction

Lifesaving is the application of skills to rescuing and reviving people in danger, especially those in water. To perform an effective rescue, lifesavers demonstrate rescue and first-aid skills that require speed and fitness [1]. While lifesaving demands high standards of physical and mental agility, it has gradually grown into a competitive sport. 

Global recognition of this sport began when the first Worldwide Lifesaving Competition took place in London in 1985 [2]. In Hong Kong, lifesaving has been recognized as a high-performance sport since early 2000s [3]. Despite its short history, lifesaving sport developed rapidly, both internationally and locally. Hong Kong representatives won their first gold medal in the Lifesaving World Championships in 2016, in which Hong Kong was ranked in the top 20 [4]. This type of competition offers an acknowledgement of lifesaving athletes’ performance at both national and international arena level. Hence, demand for scientific research to provide evidence-based physiological mechanisms behind athletes’ performance is expected in the near future.

Lifesaving sport bears a resemblance to swimming, requiring participants to demonstrate the ability to swim [5]. It involves events which mimic rescue techniques to carry drowned bodies in real life situations. Common lifesaving sport events include the manikin carry, rescue medley, and rope throw. Among these events, manikin carry is utilized as one of the most basic techniques. During the manikin carry, athletes propel themselves with one arm stroking and the other arm stabilizing a 60-kg manikin next to their body [6]. One common method of stabilization is the “Back-of-neck carry”, in which the athlete uses a firm backhand grip on the manikin’s neck to safeguard it in a secured position [7].

Despite increasing recognition of lifesaving sport, studies about lifesaving athletes’ performance and health are scarce. The very few studies about this elite sport were conducted by Abraldes et al. [8,9,10], who investigated the influence of using different fin types on athletes’ speed in 25-m manikin carry competitions. However, no underlying physiological mechanisms behind their performance have been explored. The application of dynamic integrated sEMG and motion analysis has provided objective and quantifiable data for better understanding of the intricate muscular interrelationships during athletic activities [11] in numerous sports, including throwing, tennis, and swimming. Glousman [12] found that the role of each shoulder muscle can be function-specific, even during the same sports event. Knowing the muscles’ activation profile according to their individual mechanical quality therefore formulates a sound scientific ground for more goal-oriented training.

Surface electromyography (sEMG) has been proven in many studies to be a reliable method with which to measure muscle activities during aquatic exercises [13]. However, the measurement of muscle activity underwater involves more complicated procedures than when it is performed on land. In the past, to keep the sensors functional and retrieve the data underwater, it required significant preparation, such as the application of liquid bandages on electrodes prior to the experiment [14]. Nevertheless, the development of wireless waterproof sEMG sensors, as in this study, provides a perfect solution, since it can simplify the procedure of measuring muscle activity underwater.

To identify a meaningful contraction, Jonsson [15] found that in workplace settings, the EMG force relationships between the shoulder muscles are linear only up to 30% of their maximal voluntary isometric contraction (MVIC), indicating 30% MVIC as an indicator to define significant muscle fiber activation. Moreover, handgrip strength and anthropometric properties, such as first web space, were found to have a good correlation with the time performance of elite athletes [16]. These factors might make a contribution to lifesaving athletes’ manikin-carrying arm; therefore, they should be considered when investigating the muscle activation profile and the relationship between muscle activities and swimming speed. 

With the aim of providing a directional and scientific ground for future lifesaving sports development, this study was designed to investigate shoulder muscle activity using sEMG and other correlating factors related to muscle activation. These findings can therefore facilitate a better understanding of the underlying physiological mechanism behind their performance and provide coaches with recommendations on designs and modifications of training protocols for enhancing athletes’ performance. It is hypothesized that (1) all the shoulder muscles related to the manikin carry are significantly activated throughout the 25-m swim, and (2) that the muscle activity of the selected shoulder muscles is related to speed, anthropometry, and swimming skills.

## 2. Materials and Methods

Twenty young lifesaving athletes were recruited from the Hong Kong Lifesaving Society’s elite and junior teams. Subjects were included if they (a) were aged 12 or above; (b) had at least 1 year of lifesaving sport training experience in Hong Kong; (c) had 50-m manikin carry as one of the regular training/competition events. Subjects were excluded if they had (a) history of shoulder, hip, knee or spine surgery; (b) current complaint of symptoms in any body region; (c) scoliosis; (d) regular medications; (e) chronic heart and lung disease; (f) skin allergy and/or untreated open wound; (g) any other self-/coach-reported important medical condition that may raise health concerns over participation.

This research was a cross-sectional exploratory study. Prior to the commencement of data collection, the study was approved by the Hong Kong Polytechnic University Ethics Committee (Reference Number: HSEARS20180721001) and written informed consent was signed by all athletes before participation.

There are various styles of manikin-carrying. In this study, we limited the subjects to performing the “back-of-neck carry” style, in which the swimmer uses a firm backhand grip at the back of the manikin’s neck to safeguard the victim in a secured position [7], to reduce variations. To stabilize a 60-kg, fully-filled manikin close to the body, shoulder muscles have to work against two forces (Figure 1), namely (1) gravity acting on the manikin, and (2) caudal pull from water turbulence against forward propulsion. Sustained internal shoulder rotation, along with the extension and adduction of the manikin-carrying arm, were commonly found in all athletes, regardless of the carrying angle of their shoulder and elbow. Internal rotation of the arm assisted in holding the back of the neck of the manikin; shoulder extension served to work against gravity while shoulder adduction maintained the manikin close to the athlete’s body. 

A preliminary EMG study was performed, whose objective was to identify specific muscles with meaningful activation in terms of % maximum voluntary isometric contraction (i.e., at least 30% MVIC) among the seven chosen muscles by means of the sEMG system. 

The subjects underwent a total of three trials of standardized MVIC test on each specific muscle by the same tester in a standardized position [17,18,19,20], according to the procedures showed in Table 1. The electrode positions are shown in Table 2. 

The five subjects were asked to have one trial of the 25-m manikin-carry at their maximum exertion after the MVIC of the seven muscles. They were asked to perform the manikin-carry trial in the “back-of-neck carry” style.

The sEMG results suggested that the athletes showed meaningful activation in the middle trapezius (MT), teres major (TM), posterior deltoid (PD), and latissimus dorsi (LD), in descending order (Table 3). 

After a preliminary study, the current study involved twenty athletes and was implemented in the Michael Clinton swimming pool at the Hong Kong Polytechnic University, with a water temperature within the range of 25–28 °C (±1 °C). Three electrode sensors were applied with the wireless sEMG system, PD, MT, and TM, and the highest muscle activation levels (>40% of MVIC) were examined. Since, compared to the three selected muscles, LD is a shoulder prime mover instead of a stabilizer and demonstrated lowest average % of MVIC (39.63%) among the four muscles, it was eventually excluded from the study. The objective was to use the wireless sEMG system to investigate the muscle activity of the three selected muscles and their relationship with several parameters, denoted below. 

The speed of the athletes was calculated over a 25-m distance divided by the number of seconds (two decimal places) counted by the same tester by reviewing videos. The number of inhalations was quantified by counting during a video review.

### 2.1. Procedures in Study

#### 2.1.1. Electrode Attachment

Wireless waterproof EMG electrodes (Myon, Cometa Systems, Milan, Italy) (Figure 2a) were attached onto the PD, TM, and MT of the manikin-carrying arm, in accordance with the standardized position. Special rubber gaskets (Figure 2b) were applied to affix the clip (Figure 2c,) which bore a pre-gelled surface on the opposite side to the electrode connection. The waterproof body was secured onto the skin surface by double-sided tape. (Figure 2d). The attachment locations of the EMG electrodes followed the guidelines of SENIAM (surface electromyographhy for the non-invasive assessment of muscles) [18].

#### 2.1.2. Data Recording

The subjects were requested to perform all-out 25 m manikin-carrying. During the whole testing procedures, the EMG signals were recorded at a sampling frequency of 2000 Hz and amplified by a gain factor of 1000. Other parameters included an input impedance of >10^7^ Ω, a common mode rejection ratio of >120 dB, and a bandwidth of 10 Hz–500 Hz. The EMG signals were then detected by a Wave Plus receiver (Cometa, Milano, Italy) connected to the computer. The remote acquisition mode in Data Acquisition Tools (DAT) software (Cometa, Milano, Italy) was used for wireless transmission within a distance of 20 m. The signals acquired were exported as ASCII format into a .c3d file to be analyzed.

#### 2.1.3. Grip Strength Measurement

Grip strength was measured by the same tester using a Jamar Hydraulic Hand Dynamometer. Measurement was conducted in standardized position, where subjects were seated with back support, upper arm at the side of the body, elbow at 90 degrees, and forearm in mid-pronation. Three trials were performed, and the average grip strength was calculated.

#### 2.1.4. Anthropometric Measurement

The anthropometric characteristics were measured. The lengths of the first web space and the upper and forearm (i.e., lever arm from the point of muscle exertion to the point of loading force) were recorded by the same tester using measuring tape. First web space was measured in full thumb abduction (Figure 3a). Upper arm length was measured from the greater tuberosity of the humerus to the ulnar styloid with the elbow fully extended (Figure 3b), while forearm length was measured from the olecranon process of the ulna to the radial styloid (Figure 3c). 

### 2.2. Data Analysis

#### 2.2.1. Data Processing

Raw sEMG signals were processed by bandpass filter (at 20 to 300 Hz) and root-mean-square sliding window (50 ms time constant) (MatLab2017a, Mathematical computing software, Natick, MA, USA). The average muscle activity of each muscle was normalized using its peak 1-s root mean square value, so the average was expressed as a percentage of the MVIC (%MVIC). The %MVIC were then compared. EMG data and speed in the self-pacing manikin-carrying process were further processed. The whole swimming procedure was divided into and analyzed in three phases: initial, middle, and end phase. Initial phase was defined as the period from athlete’s first swimming stroke to the end of the third stroke; middle phase was defined as the period between the initial and end phase; end phase was defined as the period from the last third stroke to the last stroke at the 25-m finishing line.

#### 2.2.2. Statistical Analysis

IBM SPSS statistics (Version 25) was used to conduct statistical analysis. Shapiro–Wilk tests were used to examine the normality of the variables and Levene’s tests were used to check the homogeneity of the variances. To evaluate between-group differences of muscle activation in different manikin carrying styles and in athletes with and without past shoulder injury, independent t-tests were used for normally distributed and homogenous data; otherwise, Mann–Whitney U tests were used. For differences in muscle activation in different phases within the same subjects, one-way repeated measures ANOVA was used, and significant comparisons were indicated by Bonferroni post hoc tests. To quantify associations between muscle activation (dependent variable) and anthropometry, speed, grip strength, number of inhalation (independent variables), Pearson’s r and Spearman’s rho were used for parametric and non-parametric data, respectively. The strength of correlation was classified into little or none (correlation coefficient (r) < 0.25), fair (r = 0.25–0.49), moderate-to-good (r = 0.5–75), and good-to-excellent (r > 0.75) (Portnet & Watkins, 2007). The level of significance was set at 0.05 for all analyses.

## 3. Results

### 3.1. Subject Characteristics

The age of the participants ranged from 13 to 23 years old with a mean of 17.75 (standard deviation (SD) = 2.29). Their life-saving training experience ranged from 1 to 13 years (mean = 2.73; SD = 1.53). Ten of the participants were female and ten were male (Table 4). 

### 3.2. Muscle Activation throughout the Manikin-Carrying Swimming

All the muscles studied had an activation of higher than 30%MVIC in manikin carrying (Table 5). Throughout the three phases, the mean of % MVIC of PD and MT declined, with the exception of TM, which had increased activation in the middle phase and maintained a consistent level of activation throughout until the end phase.

Using statistical analysis of one-way repeated ANOVA across time, the mean activation of both PD and MT in the initial phase were found to be significantly greater than in the middle and end phases, but no significant difference was found for the activation of TM between the initial and middle phase (Table 5).

#### 3.2.1. Initial Phase

As shown in Figure 4, the mean MVIC in MT activation was highest in the initial phase. Furthermore, a significant difference was found among the activation of the three muscles (F-value *=* 3.282, *p*-value (*p*) *=* 0.048). The muscle activity of MT was not only the highest in this phase but also significantly higher than that of PD (*p*
*=* 0.001, 95% confidence interval (95%CI): 0.113–0.463) in the post hoc test.

As shown in Table 6, it was found that the average muscle activation of PD was negatively correlated with the speed of this phase (unstandardized B coefficient (B) = −0.715, R-squared (R^2^) = 0.219, *p* = 0.038). Concerning the relationship between the anthropometric data and muscle activity (Table 6), the first web space was negatively correlated with both PD (B = −0.084, R^2^ = 0.239, *p* = 0.029) and MT activation (B = −0.11, R^2^ = 0.407, *p* = 0.002). Additionally, grip strength was negatively correlated with MT activation (B = −0.023, R^2^ = 0.358, *p* = 0.005).

#### 3.2.2. Middle Phase

As shown in Figure 5, the general muscle activity during the middle phase was more prominent in TM and MT. In a one-way repeated ANOVA within the middle phase, the activations between these three muscles were significantly different (F = 6.882, *p* = 0.007). From the post hoc pairwise comparison, the activation of PD was significantly smaller than TM (*p* = 0.04) and MT (*p* < 0.01).

#### 3.2.3. Middle Trapezius

Comparing the activities of all the muscles with the athletes’ performance, only the muscle activity of MT was negatively correlated with the speed of the athlete (B = −0.874, R^2^ = 0.491, *p* = 0.001). When comparing muscle activities and anthropometric features, MT activation was negatively related to both the first web space and the grip strength of the athletes (B = −0.096, R^2^ = 0.453, *p* = 0.001; B = −0.019, R^2^ = 0.368, *p* = 0.005) (Table 6). By contrast, MT activity was positively related to the number of inhalation (B = 0.06, R^2^ = 0.501, *p* < 0.001) (Table 6).

#### 3.2.4. Posterior Deltoid and Teres Major

The activation of PD was negatively correlated with the speed of the athlete (B = −0.475, R^2^ = 0.267, *p* = 0.020) and forearm length (B = −0.049, R^2^ = 0.241, *p* = 0.028). The activation of PD was also positively related to the number of inhalations (B = 0.039, R^2^ = 0.260, *p* = 0.022). However, there was no significant relationship between TM activation and speed, the athletes’ anthropometric features, or the number of inhalations (Table 6).

### 3.3. Elbow Style and Athletes’ Performance

Although the subjects were requested to perform the “back-of-neck carry” during the trial, it was observed that there were slight differences in their technique in terms of the range of motion of the elbow. They were classified into elbow-bent and elbow-straight groups. There was a significant difference in the EMG signal of TM in the middle phase between the two groups (Table 7). Athletes carrying the manikin with bent elbows showed greater muscle activation in TM (92.8%) than athletes with straight elbows (46.1%) (*p* = 0.047).

## 4. Discussion

This study aimed at constructing a more comprehensive shoulder muscle activation profile of lifesaving athletes during manikin carrying to provide answers for two topics: (1) Which muscles are activated and what are their roles throughout manikin carrying; and (2) what factors affect muscle activities and athletes’ performance. Eventually, this study revealed the three most activated muscles, their respective roles with respect to different phases of manikin carrying, as well as some positive and negative related factors that predict the activation of these muscles and the swimming speed of athletes.

### 4.1. Muscle Activation across All Three Phases

It was found that the posterior deltoid (PD), middle trapezius (MT), and teres major (TM) are the three most important shoulder muscles for performing manikin carrying as they were the only muscles with significant activation or, in other words, 30% MVIC or above, throughout the swim. All three muscles sustained significant activation throughout the 25-m swim. However, in contrast to our understanding that muscle activation increases upon exertion, a general decreasing trend in muscle activation was observed from the initial to the end phase. Two possible reasons behind this include a voluntary decrease in effort when athletes approach the finishing line, or a possibility of central fatigue phenomenon in maximal effort. Taylor et al. [21] proposed that in sustained maximal contractions, there is initially a full recruitment of the motor neuron pool. However, upon recovery from repeated excitation, our motor units become less responsive to synaptic signals. Repeated inputs from muscle afferents also produce a recurrent inhibition of signal transmission. Therefore, the descending drive from the central nervous system declines. This central fatigue mechanism occurs even in a weak contraction with less than 15% MVIC, reducing both motor unit firing and force output.

A further comparison between the muscle activation levels at different phases helped our understanding of their respective roles during manikin carrying. Although all three muscles sustained significant activation (i.e., >30% MVIC) throughout all phases of the swim, a decreasing trend was only observed in the mean activities of the PD and MT, particularly from the initial to the middle phases. By contrast, TM showed a consistent level of EMG signals, except for the last three strokes, suggesting its role in providing enduring stabilizing force to the manikin’s shoulder across time, regardless of swimming speed.

### 4.2. Initial Phase

#### 4.2.1. Muscle Activation in Initial Phase

From the investigation into the muscle activity during the initial phase, the muscle activation of MT was the highest and significantly higher than PD in terms of mean % MVIC. This finding suggests that MT could be used to a greater extent than PD to initiate the swim by elevating the shoulder girdle and, thus, the manikin towards the water surface. 

#### 4.2.2. Muscle Activation and Speed in Initial Phase

Although MT was found to have the highest activation, we found that only the activation of PD was negatively correlated with the speed in this phase. One of the possible explanations for this is that with the neck of manikin held by the athlete as an anchor point, there was a reaction force of push-off acting on the front of the manikin during this phase, providing an uplifting torque to the manikin. With higher speeds providing more uplifting torque to the manikin, the effort required to bring the manikin to water surface reduced and, thus, decreased the activation of the shoulder extensors, which were PD in our study. 

#### 4.2.3. Muscle Activation and Anthropometry in Initial Phase

A negative correlation was found between anthropometry and muscle activation in initial phase, suggesting that athletes with larger first web space activated significantly less of their PD and MT. With larger web space, it was more likely that the athlete could hold the head of the manikin tighter and exert a greater pull on the manikin efficiently with higher speed and uplifting torque. As a result, the requirement of PD and MT activity to elevate the manikin to water surface reduced. Another result showing the muscle activation of MT was smaller for athletes with stronger handgrip, possibly for the same reason.

### 4.3. Middle Phase

#### 4.3.1. Muscle Activation in Middle Phase

During the middle phase, the significant difference in the three muscle activities implies the importance of specific muscle groups in this phase. Upon further analysis, the significantly lower PD activation in the middle phase than in initial phase, as well as among the three muscles, indicates that the PD had a smaller role than the other muscles in the middle phase. In fact, the athletes already generated sufficient speed for buoyancy of manikin after the initial phase, so no sustained work of the PD for shoulder extension against gravity was needed. In short, it can be deduced that the MT is the prime stabilizer in the middle phase and that the TM maintains all-time consistent muscle activities throughout the swim. The MT is important for performing scapular retraction to provide proximal stabilization for the positioning of the manikin-carrying shoulder while TM provides sustained work to maintain a back-of-neck grip in internal rotation, regardless of the athlete’s speed, anthropometric features, and swimming skills, which explains why it was the only muscle that had insignificant correlations with all the aforementioned factors.

#### 4.3.2. Muscle Activation and Swimming Skills

The positive correlation between MT and PD activation and the number of inhalations may be explained by the fact that with more breathing and upper body turning, more turbulence acted on the athletes and the manikin. A higher muscle activation was required to compensate for the additional turbulence. The largest effect size of the MT shows that more inhalation meant an 8% decrease in MT activation. However, it should be noted that this relationship was not investigated in the initial phase because the effect of inhalation was limited. Indeed, most athletes do not need as much ventilation at the first three strokes. Furthermore, push-off by the lower limbs is likely a main confounder of this relationship, which our study did not address.

#### 4.3.3. Muscle Activation and Anthropometry in Middle Phase

The negative correlation between the anthropometric characteristics of athletes and the muscle activation of the MT is consistent with that in the initial phase; first web space is a more powerful factor than grip strength with muscle activity. Every 1-cm increase in first web space indicates a more than 10% decrease in the MT effort needed. This correlation further predicts that larger first web space to hold onto the manikin and greater grip strength are advantageous qualities for life-saving athletes. However, neither upper nor forearm length are significant predictors in the initial and middle phases.

### 4.4. Muscle Activation in Different Elbow Ranges of Motion in the Stabilizing Arm

The EMG data showed significant differences in the activation of TM, but not in the other two muscles, between the elbow-bent group and the elbow-straight group. In an elbow-bent position, the weight of manikin applied an external rotation torque on the carrying shoulder. In order to hold the manikin above and parallel to the water surface, TM as a shoulder internal rotator was activated to resist the torque. In the elbow-straight position, the external rotatory torque was smaller due to a shorter lever arm, and the direction of force did not favor a rotatory force. Hence, to avoid fatiguing the TM of the stabilizing arm during the “back-of-neck carry”, the straight-elbow technique is recommended.

### 4.5. Implications for Coaching

Based on the findings from the initial and middle phase, it is advised that from a performance-boosting perspective, the training priority should be focused on PD, MT, and TM with respect to their specific roles in different phases. The performance of the three muscles also serves as an important indicator of manikin-carrying performance that warrants regular evaluation. Furthermore, athletes’ first web space and grip strength are two good indicators for coaches to select potential lifesaving athletes with good physical qualities.

## 5. Limitations

We have addressed several limitations in this study, based on which we would like to provide recommendations for future studies:

### 5.1. Mathematical Analysis

The purposive sampling method was adopted. The selected small sample size of subjects mainly from one lifesaving society was not representative enough of the whole life-saving population in Hong Kong. Regarding this problem, nonparametric tests were used for bias-deviating factors. The 95% CI levels and standard deviation and median values of the data were cross-checked with the mean values. Furthermore, the interpretation of the results was based not only on the significance (i.e., *p*-value) but also on the effect size to avoid bias from outliers. For future studies with larger sample sizes available, stratified sampling with prior sample size calculation should be conducted.

### 5.2. Study Design

An intentional reduction in effort was suspected despite the standardized instruction that the athletes should swim for 25 m using maximal effort. To ensure the EMG collected reflect true maximal muscle activity, the end phase or last three strokes were trimmed down before analysis. In future studies, an athlete-blinded finishing line is advised. Utilizing automatic pressure or an infrared sensor at the finishing line can also ensure that athletes finish the designated distance with maximal effort.

Moreover, the number of strokes in each phase was small in our study, which may have affected the reliability of the data. However, since a 25-m trial, which is a relatively short distance, was used in this study, the number of repetitions was very limited when the trial was divided into three phases. Therefore, a longer trial, such as a 50-metertrial, should be used in future studies in order to improve the reliability of the data.

### 5.3. Subjects

In our study, the variation of training years and age was huge among our subjects, which may have reduced the significance of the results. However, we have a very limited number of competitive lifesaving athletes in Hong Kong, which limited the homogeneity of the subjects in this study. In future studies, the inclusion criteria should be set more precisely in terms of experience and age to reduce the training effects on the results.

This study serves as a pioneer study for the investigation of muscle activation in life-saving sports. It provides future directions for extended studies on other related body parts, such as the core and lower limb muscles. In addition to electromyography, other parameters, such as accelerometry, can be investigated for a more comprehensive understanding of the underlying physiological mechanism. Lastly, other study designs, such as prospective cohort studies and RCTs, can be adopted to design and evaluate the effectiveness of performance-enhancing programs. This could eventually help to formulate a sound scientific ground for more goal-oriented training for elite lifesaving sport.

## 6. Conclusions

This study is, to our knowledge, the first study to investigate the activation of the shoulder muscles during manikin carrying in lifesaving sport. The PD, MT, and TM are significantly used, each with specific roles, in manikin-carrying. According to our findings, the role of the TM was to provide an enduring stabilizing force during the initial and middle phase; meanwhile, the MT was the prime stabilizer, provide proximal stabilization for the positioning of the manikin in the middle phase. For the PD, its main function was to work with the MT to elevate the manikin towards the water surface against gravity in the initial phase. The activation of these muscles was related to the velocity, skills, and anthropometry of young elite athletes. The findings showed a decrease in the activation of PD when velocity had been built up. It suggested that muscle activity increased with a greater number of inhalations in terms of skills. Furthermore, muscle activity was negatively correlated with the first web space of the subjects during the manikin carry. A training program for lifesaving athletes is suggested to focus more on the strengthening of the three shoulder muscles in respect to the different phases of swimming. This study provides a reference for coaches in terms of athlete selection and skill training direction with regards to anthropometrics properties, i.e., hand grip strength and first web space.

## 7. Practical Implications

The muscle activity of the posterior deltoid, middle trapezius, and teres major muscles is of the highest importance for the manikin-carrying lifesaving sport event.

Coordination and strength training should be focused on these three shoulder muscles during the training session for manikin-carrying athletes. Anthropometry measurements. including first web space and upper limb grip strength, should be integrated in the athletes’ selection and identification. 

## Figures and Tables

**Figure 1 sensors-22-02143-f001:**
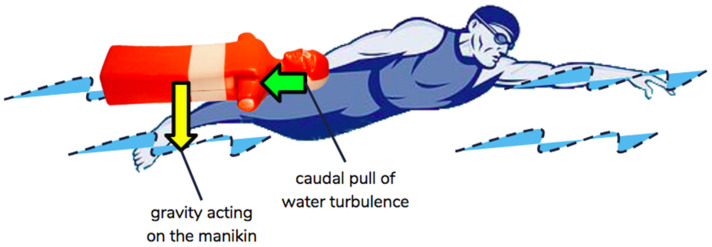
Two vector forces on the manikin during manikin carrying. Yellow arrow represents gravitational pull and green arrow represents caudal pull from the water turbulence.

**Figure 2 sensors-22-02143-f002:**
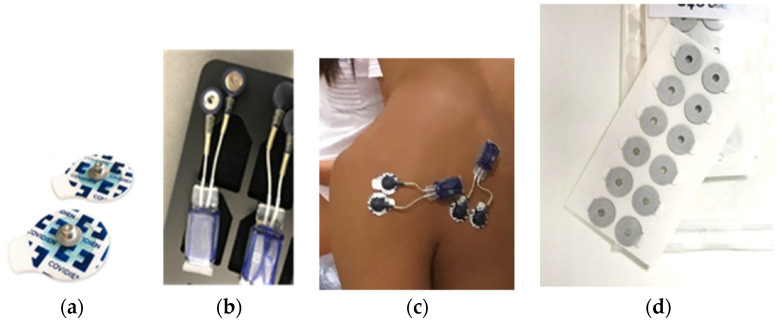
Electrode attachment: (**a**) wireless EMG electrodes; (**b**) rubber gaskets; (**c**) clips with pre-gelled surfaces; (**d**) double-sided tape.

**Figure 3 sensors-22-02143-f003:**
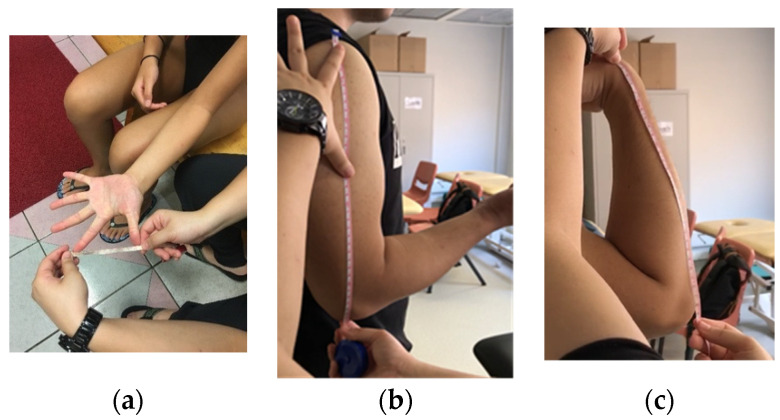
Methods of measurement for: (**a**) First web space; (**b**) upper arm length; (**c**) forearm length.

**Figure 4 sensors-22-02143-f004:**
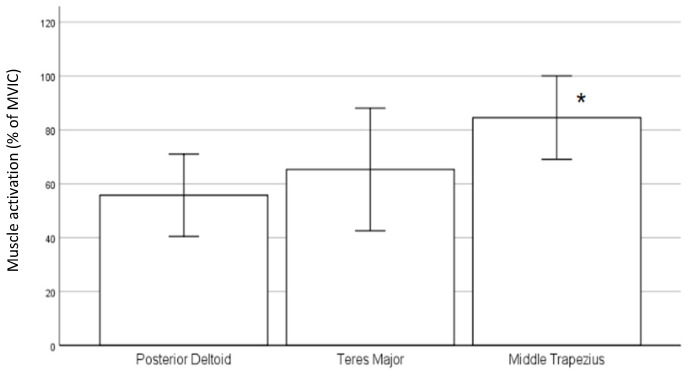
Muscle activity in initial phase (in % of MVIC). * Significantly higher muscle activation of middle trapezius compared to posterior deltoid (*p* < 0.05).

**Figure 5 sensors-22-02143-f005:**
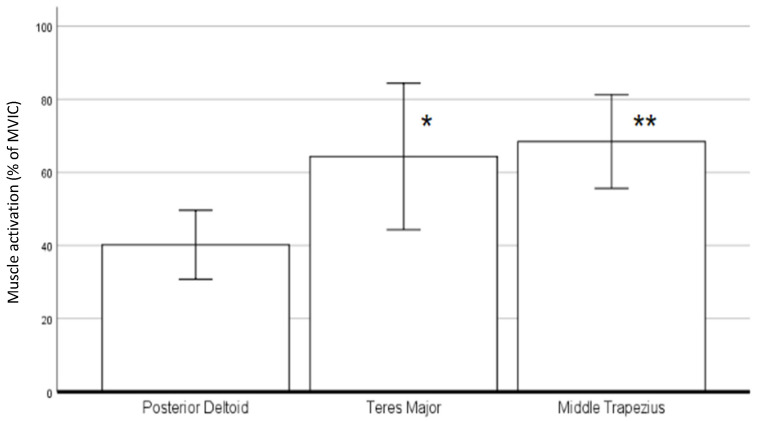
Muscle activity in middle phase (in % of MVIC). * Significantly higher muscle activation than posterior deltoid (*p* = 0.04). ** Significantly higher muscle activation than posterior deltoid (*p* < 0.01).

**Table 1 sensors-22-02143-t001:** Procedures of MVIC tests of the seven shoulder muscles among five subjects from preliminary study.

Muscle	Position	
Anterior Deltoid	The subject is seated. The arm is at the side, with the shoulder in slight abduction and the palm facing medially, with the elbow flexed at 90 degrees. Stabilization is provided via the scapula and clavicle. Resistance is applied on the anteromedial aspect of the arm just proximal to the elbow joint, in the direction of shoulder extension, slight abduction, and external rotation.	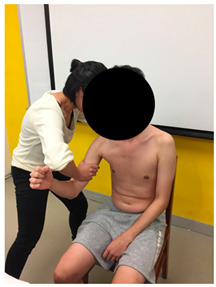
Middle Deltoid	The subject is seated. The test arm is at the side in neutral rotation, and the elbow is flexed 90 degrees. Stabilization is provided via the scapula. Resistance is applied proximal to the elbow joint on the lateral aspect of the arm in the direction of shoulder adduction.	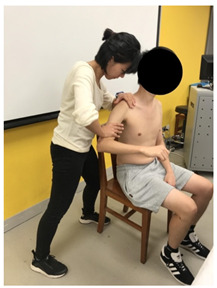
Posterior Deltoid	The subject is seated, with the shoulder abducted, and elbow flexed at 90 degrees. The shoulder is kept at 30 degrees of internal rotation (i.e., forearm pointing downwards). The subject is requested to perform an isometric contraction of shoulder extension.	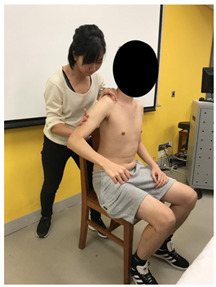
Latissimus Dorsi	The subject is lying prone, with the shoulder extended 20 degrees; the forearm is pronated and adducted with the hand positioned on the ipsilateral buttock. Stabilization is provided via the ipsilateral scapula. Resistance is placed over the forearm and is directed towards flexion and abduction.	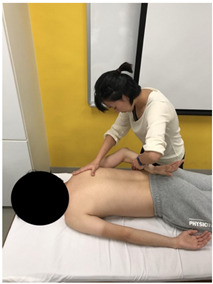
Middle Trapezius	The subject is lying prone, with the shoulder at the edge of the table, abducted to 90 degrees; the elbow is flexed at a right angle. Stabilization is provided via the contralateral scapula. Resistance is placed over the distal end of the humerus during scapular retraction and directed downward toward the floor.	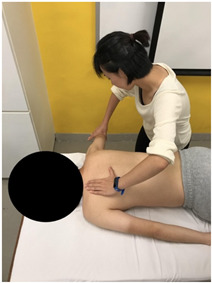
Teres Major	The subject is erect, sitting with the testing arm abducted 45 degrees and internally rotated 30 degrees. Stabilization is provided on the distal forearm and externally rotated.	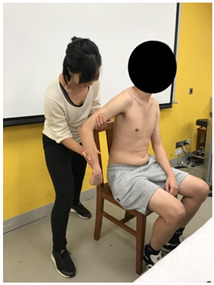
Pectoralis Major	The subject is seated. The shoulder is abducted to 90 degrees, and the elbow is flexed to 90 degrees. Stabilization is provided by the weight of the subject’s weight, and via the contralateral shoulder. Resistance is applied on the anterior aspect of the arm proximal to the elbow joint in the direction of shoulder horizontal abduction.	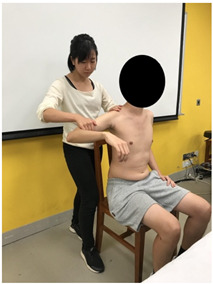

A 2-min recovery period was allowed between trials to reduce the effects of fatigue.

**Table 2 sensors-22-02143-t002:** Electrode Positioning.

Muscle	Location of Electrode	Orientation of Electrode
Anterior Deltoid	At one-finger-width distal and anterior to the acromion	In the direction of the line between the acromion and the thumb
Middle Deltoid	From the acromion to the lateral epicondyle of the elbow, corresponding to the greatest bulge of the muscle	In the direction of the line between the acromion and the hand
Posterior Deltoid	Center the electrodes in the area about two finger breadths behind the angle of the acromion	In the direction of the line between the acromion and the little finger.
Middle Trapezius	4 cm lateral to the spinous process of T3	In the direction of the line between T5 and the acromion.
Latissimus Dorsi	Over the muscle belly at the Tl2 level	Along a line connecting the most superior point of the posterior axillary fold and the S2 spinous process
Teres Major	2 cm lateral to the inferior angle of scapula	Parallel to muscle fibers going laterally up from scapula angle to the arm
Pectoralis Major	5 cm below the most medial point of the clavicle	Parallel to muscle fibers horizontally

**Table 3 sensors-22-02143-t003:** Muscle activation (% of MVIC) of the seven shoulder muscles among five subjects from preliminary study.

Muscle	Subject 1	Subject 2	Subject 3	Subject 4	Subject 5	Average
Anterior Deltoid (AD)	8.10	6.57	3.847	13.47	11.33	8.66
Middle Deltoid (MD)	15.74	20.38	04.78	28.25	21.20	18.07
Posterior Deltoid (PD)	30.75	51.31	39.55	43.62	47.92	42.63
Teres Major (TM)	62.32	35.62	58.60	28.77	53.85	47.83
Pectoralis Major (PM)	7.10	03.52	11.57	03.78	21.34	9.46
Latissimus Dorsi (LD)	44.49	38.46	56.90	29.51	28.80	39.63
Middle Trapezius (MT)	45.41	57.27	73.05	50.04	50.03	55.16

**Table 4 sensors-22-02143-t004:** Subject Characteristics.

Number of Subject	20
Gender	10 male; 10 female
Age (Mean ± SD)	17.75 ± 2.29
Year of Experience (Mean ± SD)	2.73 ± 1.53
Height (m) (Mean ± SD)	1.68 ± 0.07
Weight (kg) (Mean ± SD)	60.65 ± 7.71
Body Mass Index (kg/m^2^) (Mean ± SD)	21.46 ± 2.07
Upper Arm Length (cm) (Mean ± SD)	33.93 ± 1.94
Forearm Length (cm) (Mean ± SD)	27.97 ± 2.01
First Web Space (cm) (Mean ± SD)	15.20 ± 1.92
Grip Strength (kg)(Mean ± SD)	33.15 ± 8.56
History of Shoulder Symptoms	5 subjects with past history
	15 subjects without past history
Manikin carrying style	8 subjects carried with elbow bent
	12 subjects carried with elbow straight

**Table 5 sensors-22-02143-t005:** Summary of muscle activities (%MVIC) of PD, MT, and TM in three phases.

Phase	Posterior Deltoid	Middle Trapezius	Teres Major
	Mean ± SD	Mean ± SD	Mean ± SD
Initial	55.73 ± 3.59 ^a^	84.54 ± 5.56 ^b^	65.26 ± 1.84
Middle	40.21 ± 20.20	68.43 ± 27.40	64.35 ± 42.80
End	26.82 ± 5.84	52.93 ± 1.09	49.52 ± 1.08

^a^ Different from middle phase (*p*
*=* 0.005) and end phase (*p* < 0.001). ^b^ Marginally different from middle phase (*p*
*=* 0.062) and end phase (*p*
*=* 0.002).

**Table 6 sensors-22-02143-t006:** Summary of correlation between muscle activity and speed, anthropometry, and skills.

Parameter	Speed		First Web Space	Grip Strength	Upper Arm Length	Forearm Length	Number of Inhalation
	Initial Phase	Middle Phase					
Initial Phase							
Posterior Deltoid %MVIC							
Unstandardized B coefficient	−0.715	N/A	−0.084	−0.01	−0.006	−0.068	N/A
R^2^	0.219	N/A	0.239	0.074	0.001	0.174	N/A
*p*-value	0.038 *	N/A	0.029 *	0.245	0.883	0.068	N/A
Teres Major %MVIC							
Unstandardized B coefficient	−0.429	N/A	−0.006	−0.009	−0.072	−0.053	N/A
R^2^	0.036	N/A	0.001	0.025	0.083	0.047	N/A
*p*-value	0.424	N/A	0.915	0.503	0.218	0.895	N/A
Middle Trapezius %MVIC							
Unstandardized B coefficient	−0.465	N/A	−0.11	−0.023	−0.022	−0.058	N/A
R^2^	0.091	N/A	0.407	0.358	0.017	0.124	N/A
*p*-value	0.197	N/A	0.002 *	0.005 *	0.582	0.127	N/A
Middle Phase							
Posterior Deltoid %MVIC							
Unstandardized B coefficient	N/A	−0.475	−0.046	−0.006	−0.009	−0.049	0.039
R^2^	N/A	0.267	0.433	0.064	0.007	0.241	0.260
*p*-value	N/A	0.020 *	0.056	0.283	0.719	0.028 *	0.022 *
Teres Major %MVIC							
Unstandardized B coefficient	N/A	−0.176	−0.057	−0.009	−0.067	−0.043	0.022
R^2^	N/A	0.008	0.065	0.036	0.093	0.042	0.017
*p*-value	N/A	0.706	0.276	0.425	0.192	0.389	0.582
Middle Trapezius %MVIC							
Unstandardized B coefficient	N/A	−0.874	−0.096	−0.019	−0.038	−0.041	0.060
R^2^	N/A	0.491	0.453	0.368	0.071	0.093	0.501
*p*-value	N/A	0.001 *	0.001*	0.005 *	0.255	0.192	0.000 **

* *p* value < 0.05. ** *p* value < 0.001.

**Table 7 sensors-22-02143-t007:** Muscle activation in athletes with different elbow styles.

	Elbow Bent Group	Elbow Straight Group	*p* Value
Posterior Deltoid %MVIC	41.3 ± 16.8	34.1 ± 19.0	0.408
Teres Major %MVIC	92.8 ± 53.3	46.1 ± 26.7	0.047 *
Middle Trapezius %MVIC	74.1 ± 34.3	58.6 ± 21.1	0.224

* *p* < 0.05.

## Data Availability

The data presented in this study are available on request from the corresponding author. The data are not publicly available due to data privacy.

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
