# Peer review of "Investigation of Underwater Shoulder Muscle Activity during Manikin-Carrying in Young Elite Lifesaving Athletes"

_sensors, 2022, doi:10.3390/s22062143_

Round 1

Reviewer 1 Report

General

Not all abbreviations used are explained in the text (R2, F, ...). I know these are commonly used parameters, but in my opinion all abbreviations should be explained.

Is the research design appropriate?

Mainly yes. However, increasing the number of repetitions would increase the reliability of the data (e.g. 5 repetitions of the swimming pool for each player).

row (16): When reading the abstract, the reader does not know what initial and middle phase is. So how is he to understand the results?

(115) Were EMG measurements of all seven muscles performed in parallel? If not, there are reasons for factors that should be included in applications.

(118)  Ther is no information on the MVIC procedure. The test to determine the MVIC for specific muscles should be described. It may be useful to other scientists as information on how to perform it for those specific muscles.

Additionally, it will dispel doubts as to its correct execution.

How was the MVIC calculated (conditions, equation, based on rms / mean or other coefficient)? What was the size of time window? The window could last 10ms or 100ms and these were completely different values.

Table 1. I suggest that you also include abbreviations for the names of muscles in the table.

(130) Earlier (row 120), four muscles were mentioned. Here, only three were selected. What was the selection / rejection criterion?

(143) Redundant full stop between (type and Figure 2d).

(220) I assume that not all groups were characterized by normal distribution, so non-parametric tests should have been used. What type of ANOVA was performed.

(253) "…mean of average…"?

What means the first mean? And what means the second mean (average)?

I assume that the first mean is just mean of MVCI. But what the second average means  is?

Figure 4. y axis. I assume that should be MVIC instead MVC.

Table 4. Why was the END phase not included?

Figure 5. I assume that the graph presents mean values and std. standard deviation (it is not described anywhere - it applies to all graphs). If so, based on the data from Table 3, PD should be 40.23 +/- 20.2. However, in the chart, the deviation is much smaller (I estimate it at 10)

The same applies to Fig 4 where according to Table 3 std is much smaller than shown in the drawing.

(309) I assume that the p-value relates to the correlation between speed and the selected parameter (e.g. no. Of inhalation)?

What does Coefficients mean? What are the units in Table 6? Why are the results (coefficients) in Table 6 different from the corresponding data in Table 2?

(328) According to Table 1, only two muscles (MT and PD) meet this criterion (MVIC> 30%). Why was LD in the analyzes rejected and TM left behind?

Reviewer 2 Report

Thanks for this invitation to review this study. The outcomes of this study provide novel findings in attempt to enrich the current knowledge in literature. However, this study may be limited by several methodological issues. I would suggest the authors to reconsider the issues I raised for future submission.

Major comments:

  1. Study rationales to examine middle trapezius, posterior deltoid, and teres major were not fully constructed. Despite five participants of pilot study reported, the selection criteria of measuring muscle are not well established by fundamental theory. To me, according to the table 1, latissimus dorsi needs to include the investigation based on the criteria of 30% MIVC.
  2. The method to detect the EMG signals during lifesaving task is criterial issue in this study. However, such information never comes cross in the manuscript.
  3. Why the inclusion criteria include participants aged less than 18 yrs. The wide range of participants’ age from 13 to 23 yrs should be a bias of this study.
  4. The lifesaving task is not identical among the participants. 8 participants used carried with elbow bent and 12 participants carried with elbow straight. Inconsistent performance task among the participants could lead to the results are incomparable.
  5. It is unclear to use first web space and hand grip strength to evaluate the rescue performance of lifesavers. Although the authors introduced information regarding EMG-force relationship between line 69-76, the relationship between the measuring variables and lifesaving performance is not established. The authors need to justified this issue.

Minor comments

  1. Abstract, line 15-16, please specifically point out the measuring variables instead of general terms.
  2. There is no need to capitalize the testing muscles.
  3. Please remove the decimal of percentage change in MVIC.
  4. The standard process to determine the EMG locations are not reported in the manuscript.
  5. The conclusion is relatively weak and the statement is not supported by the findings.
  6. How the authors determined MIVC?

Reviewer 3 Report

Manuscript Review - Manuscript number: sensors-1563071

     The article presents interesting reports, important from the point of view of the possibility of training people to save lives in the aquatic environment.

The title encourages you to read the content of the article.

The summary is complete.

The aim and hypothesis of the publication were clearly defined.

There are some shortcomings in the Material and Methods section.

Namely, the number of puffs was quantified by counting by one tester during the video review. It seems, however, that one counting during a video projection may not be accurate enough to be considered objective. Therefore, in similar future research, it is worth planning a more precise form of counting the results by watching a video, as well as paying attention to a better selection of participants for the study. The study participants vary greatly in terms of training years (1-13 years), which may reduce the significance of the results.

Moreover, as the authors write, the electrodes were attached "in accordance with a standardized position". What does the standardized position mean, by whom/what, the electrode application method for the surface electromyography reading, was used? This should be completed.

The tables are easy to read.

Results section - the research results confirm the assumptions of the work.

The discussion relates to the results.

The main problem that slightly reduces the correctness of the presented work is the shortcomings described above as well as the number of respondents emphasized by the authors.

Please complete the Material and Method section.

The work requires corrections before it is allowed to be published.

Round 2

Reviewer 2 Report

After reading the revision, I do not think the authors sufficiently responded the issues regarding methodological considerations. Therefore, I can not support the currently form of study for further review.